# Continuous, long-term crawling behavior characterized by a robotic transport system

**James Yu[1], Stephanie Dancausse[2], Maria Paz[1], Tolu Faderin[1], Melissa Gaviria[2], Joseph W Shomar[2], Dave Zucker[3], Vivek Venkatachalam[1]\*, Mason Klein[2]\***

[1]Department of Physics, Northeastern University, Boston, United States; [2]Department of Physics and Department of Biology, University of Miami, Coral Gables, United States; [3]FlySorter, LLC, Seattle, United States

**Abstract** Detailed descriptions of behavior provide critical insight into the structure and function of nervous systems. In *Drosophila* larvae and many other systems, short behavioral experiments have been successful in characterizing rapid responses to a range of stimuli at the population level. However, the lack of long-term continuous observation makes it difficult to dissect comprehensive behavioral dynamics of individual animals and how behavior (and therefore the nervous system) develops over time. To allow for long-term continuous observations in individual fly larvae, we have engineered a robotic instrument that automatically tracks and transports larvae throughout an arena. The flexibility and reliability of its design enables controlled stimulus delivery and continuous measurement over developmental time scales, yielding an unprecedented level of detailed locomotion data. We utilize the new system's capabilities to perform continuous observation of exploratory search behavior over a duration of 6 hr with and without a thermal gradient present, and in a single larva for over 30 hr. Long-term free-roaming behavior and analogous short-term experiments show similar dynamics that take place at the beginning of each experiment. Finally, characterization of larval thermotaxis in individuals reveals a bimodal distribution in navigation efficiency, identifying distinct phenotypes that are obfuscated when only analyzing population averages.

**\*For correspondence:**
v.venkatachalam@northeastern.edu (VV);
klein@miami.edu (MK)

## Editor's evaluation

This study describes a useful method to monitor the behavior of *Drosophila* larvae in a uniform environment over much longer time scales than was possible with previous methods. The authors provide a solid characterization of aspects of the method and show that the behavior of single larvae can be quantified over several hours. The experiments offer a proof-of-concept for a robotic device that will enable the investigation of behavior in long-term experiments in ways that were previously unimaginable.

## Introduction

A complete description of an organism's behavior, or any responsive system more generally, would include a map of how inputs transform into outputs. Reflexes or decisions made by the peripheral and central nervous systems, for example, can be characterized as functions that take surrounding environmental (and internal) stimulus information, process it, and lead to a physical behavior. An organism's transformation properties are rarely constant, and instead change over short and long time scales, determined by its stimulus history and development. A comprehensive understanding of animal behavior and its underlying mechanisms would ideally address all time scales between fast

neural responses through the slow physiological changes associated with development. Short-term responses to individual stimuli have been characterized in many organisms, but a high-bandwidth treatment with continuous measurement of behavior through the entire time course of an animal's development has been prohibitive. The lack of continuous observation makes it difficult to dissect individual behavioral dynamics and their development over time. In most organisms, behavior is too fast, too complicated, is performed in too large of a space, or otherwise too difficult to observe with sufficient resolution over a long time.

Recent work in several model systems has begun to dramatically increase the duration of behavioral recording. Roaming and dwelling behavior in worms has been recorded with high spatiotemporal resolution through larval development by via isolation of individuals in small arenas (*Stern et al., 2017*). The multi-camera imaging system has observed changes in roaming behaviors across and within developmental stages in the same animal along with providing insight into the variation of these behaviors between individuals in a population. Long-term home-cage behavior in both mice (*Goulding et al., 2008*) and rats (*Poddar et al., 2013*) has been studied using a combination of behavior-specific sensors and cameras. The home-cage systems include automated monitoring and training of multiple animals in parallel, increasing overall throughput and minimizing the human assistance required for such tasks, and have been successful in elucidating new determinants in behaviors across multiple time scales, including feeding, daily activity patterns, and learning novel motor tasks. Finally, long-duration recording of 1D walking in adult fruit flies (*Drosophila melanogaster*) has been recorded using a simple photobeam interrupt (*Pfeiffenberger et al., 2010*), and more complex behaviors have been recorded using video monitoring (*Geissmann et al., 2017*) and robotic manipulation (*Alisch et al., 2018*). Our goal in the present work is to investigate long-term locomotion in a free, open physical space, by continuously monitoring a simple model organism with a short life cycle, which will allow us to characterize exploratory search and directed navigation over a sizable fraction of the organism's life cycle.

The *D. melanogaster* larva model system (hereafter referred to as *Drosophila* or larva) presents an opportunity to study navigational behavioral dynamics over long time scales, and is well suited to a detailed quantitative treatment. It possesses a well-defined, slow behavioral repertoire and robust response to many stimuli (*Sokolowski, 1980*, *Louis and de Polavieja, 2017*, *Riedl and Louis, 2012*). The fruit fly has a relatively short life cycle with high accessibility to their three short larval stages (*Fernández-Moreno et al., 2007*). During these stages, larvae are highly food-motivated and thus in the absence of food engage in nearly continuous exploratory search movement, which facilitates behavioral studies of locomotion over long times (*Wosniack et al., 2022*). They also demonstrate responses to chemosensory cues (*Schumann et al., 2021*, *Kim et al., 2017*, *Vogt et al., 2021*, *Colomb et al., 2007*), mechanical and nociceptive stimulation (*Almeida-Carvalho et al., 2017*, *Almeida-Oliveira et al., 2021*, *Kudow et al., 2019*), light (*Kane et al., 2013*), as well as the ability to retain memory and change their behaviors in accordance with learned experiences, and habituate to prolonged stimuli (*McGuire et al., 2003*; *McGuire et al., 2004*, *Larkin et al., 2015*, *Berne et al., 2021*). Larvae also perform robust behavioral responses to temperature changes, engaging in thermotaxis along thermal gradients, and modulating aspects of their locomotive behavior, in particular their turning rate, in order to reach optimal conditions (*Luo et al., 2010*, *Klein et al., 2015*). The recent availability of a connectome brain wiring diagram (*Huser et al., 2012*, *Larderet et al., 2017*) and numerous genetic tools have facilitated probing the neural circuits and molecular mechanisms that underlie these behaviors (*Hernandez-Nunez et al., 2020*, *Kaun et al., 2012*, *Gepner et al., 2015*, *Sokolowski, 1980*, *Tian et al., 2009*, *Xie et al., 2018*, *Inada et al., 2011*). Here, we focus on directly observed free-moving exploratory search and navigation behavior and seek to continuously measure fly larva crawling over times scales of many hours.

Responses to a wide range of stimuli in larvae typically occur through changes in their navigation and locomotion. Their navigation behavior, when confined to flat 2D spaces, is akin to an organism-scale 2D random walk (*Berg and Brown, 1972*, *Berg, 2004*), characterized as an alternating sequence of forward crawling 'runs' and direction-altering 'turns', making the animal's behavior and response to stimuli straightforward to quantify (*Codling et al., 2008*, *Günther et al., 2016*, *Lahiri et al., 2011*, *Klein et al., 2017*; *Figure 1A*). However, larvae crawl away and typically remain at the edges of confining barriers, or climb walls or bury into a substrates (*Louis and de Polavieja, 2017*) (see *Figure 1—figure supplement 1*). Any such scenario results in the termination of exploratory search

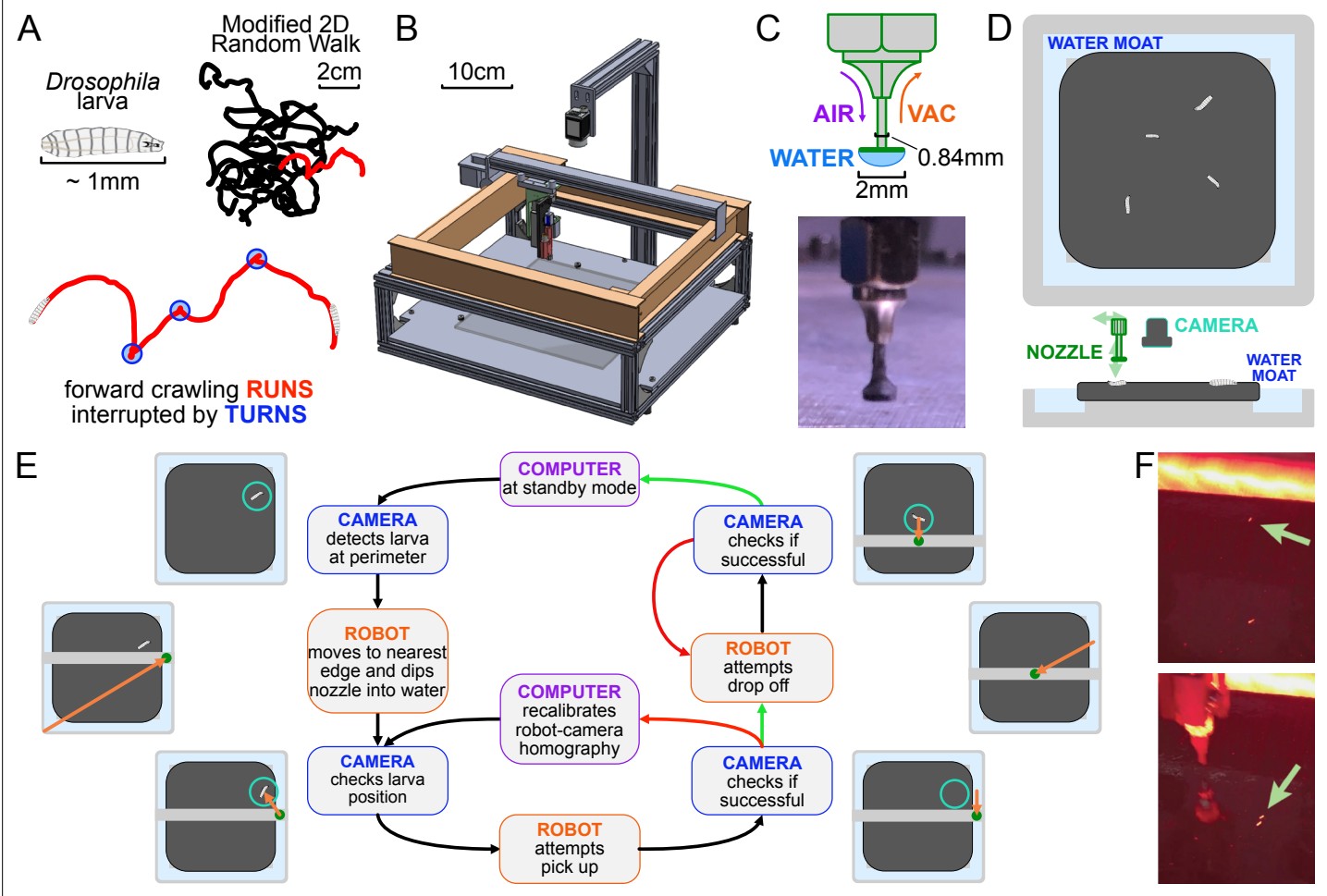

**Figure 1.** The automated larva transport robot enables continuous, long-term observation of fly larva crawling behavior. (**A**) Schematic illustrating the fly larva's simple search behavior. They search their environment in a modified 2D random walk, with 20 example paths (black) shown. Trajectories are characterized by an alternating series of forward crawling runs (red) and turns (blue). (**B**) Isometric CAD schematic of the transport robot. The robot is built from a modified 3D printer with a custom nozzle. Feedback from a mounted overhead camera allows for tight coordination with the moving arm to safely and robustly interact with the experimental arena. (**C**) The nozzle is built as a narrow tube that allows air and vacuum flow with a flat plastic disk fitted at the bottom. The disk provides ample surface area for a water droplet to form, and the droplet's surface pressure can pick up larvae while minimizing stress on the animal during the interaction. (**D**) Top and side view schematics of the flat crawling arena. Larvae crawl atop an agar substrate, which is kept hydrated by a surrounding moat. The robot nozzle picks up larvae as they approach the edge of the arena and transports them back to the center to continue their freeroaming behavior. (**E**) Flowchart of the larva pick-up feedback process. In standby mode, the camera records a video of larval behavior. When it detects a larva nearing the perimeter, it triggers the pick-up protocol for the robot. The manipulator arm moves to a point in the moat nearest to the larva and dips the nozzle in, forming a droplet at the tip to be used for pick-up. The camera provides a more recent position for the moving larva as the robot attempts a pick-up. If feedback from the camera suggests a successful pick-up, it attempts a drop-off. Otherwise, the manipulator repeats its attempt after receiving an updated larval position. Multiple failed attempts can trigger small perturbations to robot calibration parameters to allow better flexibility through reinforcement learning before continuing pick-up attempts. Similarly, the robot performs multiple drop-off attempts at the center of the arena until it receives a positive confirmation from the camera, at which point the system returns to its original standby mode. (**F**) Photographs before (top) and after (bottom) the robot moves a larva from the perimeter to the center of the arena.

The online version of this article includes the following video and figure supplement(s) for figure 1:

**Figure supplement 1.** Time lapse from a movie of 22 larvae crawling on a 60 mm agar dish.

**Figure 1—video 1.** Real-time video showing pick-up and drop-off as in panel F, three larvae transported from the edge of the arena to the center.
https://elifesciences.org/articles/86585/figures#fig1video1

behavior, limiting most experiments to shorter snapshots of larval behavior, typically on the order of 10 min (*Jaime et al., 2018*, *Klein et al., 2015*, *Alisch et al., 2018*, *Klein et al., 2017*). A similar issue arises with directed navigation, for example along a 1D stimulus gradient, where larvae reach their preferred stimulus location or the edge of the arena after a short time, after which they no longer

navigate toward anything. Thus, we desire a system that can prolong the behavior under investigation. Manually prolonging exploratory crawling behavior, such as picking up a larva with a wet paintbrush and placing them back at the center, is inefficient, difficult to perform consistently, and too labor-intensive over long times, and thus not very practical. Longer experiments with adult *Drosophila* have successfully been automated to allow higher throughput, although with confinement (*Jaime et al., 2018*, *Alisch et al., 2018*, *Buchanan et al., 2015*), but no such system has previously been developed for freely crawling larvae.

Here, we present the design and operation of an automated 'larva picker' robot and demonstrate its capabilities through continuous observation of larval exploratory searching and navigation behavioral response with high detail and over unprecedented duration. Importantly, identity is maintained throughout tracking, so we can characterize exploration and navigation at the population and individual animal levels together, as larvae search for food under varying hunger conditions, or negotiate variable temperature environments. In doing so, we are able to reveal new behavioral dynamics, where the animals' search strategy is modulated over hours and we can discriminate between different individual statistics that are otherwise hidden by population averages.

## Results

### A robotic system transports larvae throughout an arena

To perform long-term behavioral studies with the larva, we have designed and fabricated a robotic instrument that automatically tracks, transports, and feeds larvae throughout a large arena. The flexibility of its design enables arbitrary stimulus delivery alongside continuous measurement of behavior, yielding an unprecedented level of detailed locomotion data and a comprehensive view of locomotion strategies at the population and individual animal levels together.

The system operates by tightly coordinating video acquisition from an overhead camera with a manipulator arm capable of traversing three dimensions and a custom nozzle that can manipulate larvae (*Figure 1B*).

A small number of larvae crawl on an agar gel are illuminated by red LEDs from the side, and a camera records their activity. The camera detects when a larva nears the edge of the crawling arena, and the robot arm and nozzle pick up the larva with the aid of a water droplet and place the larva back in the center. This process is shown in *Figure 1E*.

Meanwhile, the mounted camera refreshes the image and therefore the position of the larva before and after each step of any protocol, ensuring that the process is robust to variations in crawling speed and behavior and to failed attempts, which are detected and repeated (*Figure 1E*). The pick-up/drop-off procedure is triggered at an average rate of 0.87 times per hour per animal, and is 99.8%/99.9% successful (90%/95% on the first try), highly important for the viability of long-term experiments.

Our system must also address desiccation of the agar gel substrate and animal starvation, which limit experiment times and affect behavior. To address the former, we have built an auto-replenishing water moat (*Figure 1D*) in direct contact with the gel, which then maintains its water content and physical shape. In addition, the moat also provides a convenient and readily available water source for

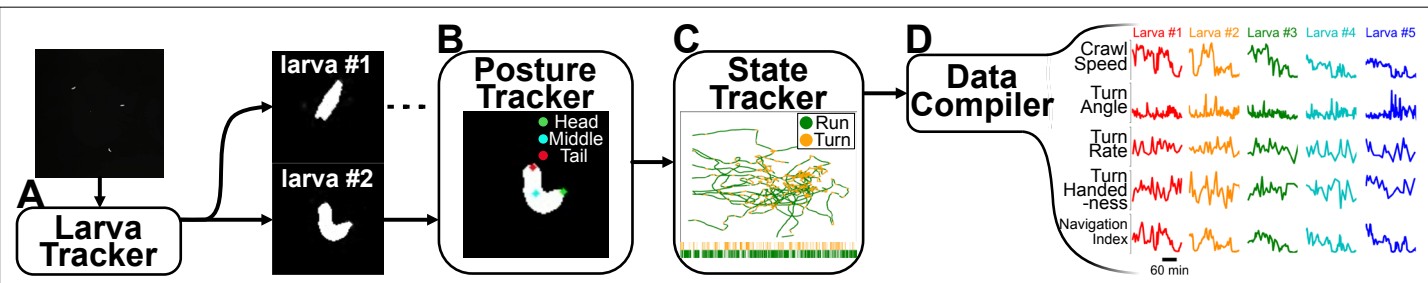

**Figure 2.** Analysis pipeline. (**A**) Raw video acquired during the experiment is fed into computer vision software that tracks each larva while maintaining their identity. (**B**) A posture tracker analyzes the isolated crops of each larva to determine its posture and orientation. (**C**) A state tracker determines the behavioral state of each animal at each time point. (**D**) Compiling all information from the preceding algorithms allows the pipeline to identify and calculate a wide variety of behavioral features.

the nozzle. To address the latter, the robot can automatically administer a drop of apple juice (≈0.1 g/mL sugar concentration) directly to the larva on a predetermined schedule. The larva is allowed to eat for a fixed time, then rinsed with water so that it can return to a free-crawling state.

With these features in place, our system has so far achieved more than 30 hr of continuous observation of individual larva behavior.

In some experiments we observe how larvae navigate a variable sensory environment. We use a similar system as outlined in *Klein et al., 2015*, to generate a 1D linear spatial thermal gradient of 0.035°C/mm with 13°C on one side and 21°C on the other.

## Video analysis identifies key behavioral features

Our data analysis pipeline extracts numerous behavioral features from video recordings of crawling larvae (*Figure 2*). Custom computer vision software takes raw movies and extracts the position and body contour of each larva while preserving individual animal identities (*Figure 2A*). Because the robot arm can briefly block the camera during pick-up events (average of ≈4.5 s of occlusion duration per pick-up event), resulting in dropped frames, the software interpolates larval positions in these frames to maintain continuous observation. Since the larva's body length is roughly 1 mm and it crawls with an average speed well below 1 mm/s, significant behavior events are unlikely to occur during dropped frames and interpolation is sufficient to rebuild trajectories.

The isolated crops of each larva are run through a recurrent U-Net *Ronneberger et al., 2015* convolutional neural network (*Figure 2B*) to determine the posture and orientation. Using all available information (position, contour, and posture), we use a densely connected neural network with bidirectional recurrence to classify the behavioral state of the larva ('run', 'turn') at each time point (*Figure 2C*; *Günther et al., 2016*). From here, we can identify a wide range of behavioral features and track them over extended time periods.

## The robot enables continuous observation of free exploratory searching over 6 hr

Optimizing searches by modulating behavior is essential to the larva's ability to find a food source efficiently. While previous studies with *Drosophila* larvae have revealed some changes to its navigation over the first few minutes of exploratory search of an isotropic environment, how and whether their behavior evolves or persists afterward remains unknown (*Klein et al., 2017*). Studies on another small organism with qualitatively comparable navigation dynamics, the nematode *Caenorhabditis elegans*, show similar changes in behavior during the first ≈ 100 s. Some longer-duration experiments (1 hr vs 15 min) reveal a transition in navigation strategy between two distinct modes of local and global searches, but transitions across similar or longer time scales have yet to be observed in *Drosophila* (*Calhoun et al., 2014*, *Klein et al., 2017*). Here, we demonstrate how our larva picker robot enables continuous observation and analysis of larval locomotive behavior on very long time scales to study changes in its search strategy in an isotropic environment.

In *Figure 3*, we present the results from continuous observation of second instar wild type (Canton-S) larvae ($N = 42$) freely roaming the experimental stage for a 6 hr time duration. *Figure 3A and C* shows the speed and the turn rate of the larvae, respectively, exhibiting the dynamics of their behavior. Notably, there is a significant drop in activity (both speed and turn rate) over the first hour before settling into a plateau that lasts for the following few hours. The correlation between larval crawl speed and turn rate over time has been previously observed (*Günther et al., 2016*) and is clearly evident here, and we measure a correlation coefficient of 0.572 in the population mean speeds and turn rates. With the large amount of data gathered on each individual, we confirm that this correlation also exists at the individual level with an average correlation coefficient of 0.348 ± 0.098.

## Long-term free-roaming behavior is consistent with analogous snapshot observations

Since the larva were not fed over the duration of the experiment, we compare continuous observations with short 'snapshot' observations of larvae at various stages of starvation ($N = 200$ per stage). Larvae were removed from food and starved over a certain number of hours, then placed on an agar gel arena to be observe for 10 min without interruption, and with no interaction with the robot. The decline in activity seen in the robot-mediated experiments is not present when observing crawl speeds

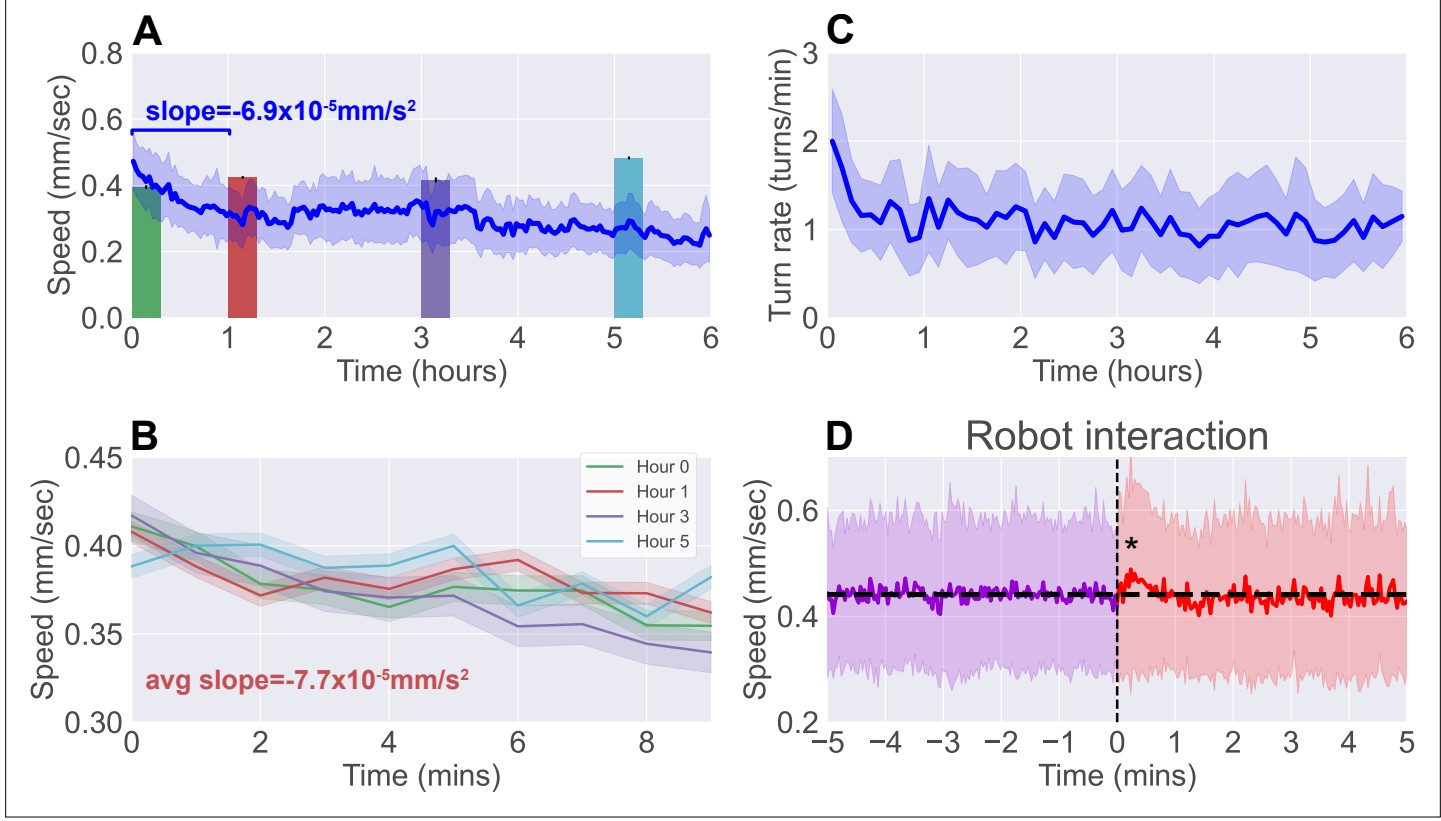

**Figure 3.** Observations of continuous free roaming for 6 hr. (**A**) Larval crawl speed over time, comparing the results from continuous 6 hr observation recorded on the robot system (blue, $N = 42$) to shorter 10 min observation of larger numbers of starved larvae (colored, $N = 200$ per bar, from 10 experiments with 20 larvae each). (**B**) Larval crawl speed over 10 min after starvation. We observe a decline in crawl speed ($-7.7 \times 10^{-5}$ mm/s²) comparable to that observed during the first hour of the continuous observation ($-6.9 \times 10^{-5}$ mm/s²). (**C**) Larval turn rate over time. Similar to larval crawl speed (**A**), there is a noticeable drop in turn rate over the first hour, indicating an overall decrease in activity. (**D**) Analysis of larval crawl speed before and after an interaction with the larva picker robot. We plot larval crawl speed during the 5 min immediately before (purple) and the 5 min immediately after (red) an interaction with the robot, that is, a pick-up and drop-off event (vertical black dashed line). After a 1–2 min transient ($p < 0.05$, Student's t-test), speed returns to the mean pre-interaction level (horizontal black line). For all panels, shaded regions indicate standard deviation from the mean.

averaged over the 10 min duration (*Figure 3A*, colored bars). Upon closer inspection, however, larval crawl speed measured in the starvation studies (*Figure 3B*) reveal a trend over time that mirrors what we observe in the first hour of continuous observations. Over these 10 min snapshot observations, we measure an average decline of $-7.7 \times 10^{-5}$ mm/s², comparable to the average of $-6.9 \times 10^{-5}$ mm/s² seen over the first hour of continuous observations. The similarity offers confidence that the behavioral dynamics present here are not caused by transport robot actions but are real features of the animal that continue to persist and develop over a period that is more than 30 times longer than what snapshot observations can capture.

To further ensure that disruptions to behavior due to the robot's interference have minimal influence, we analyzed larval behavior before and after each interaction with the robot (i.e. a pick-up and drop-off event). *Figure 3D* shows averaged larval crawl speeds 5 min before and 5 min after an interaction event (vertical dashed line). As expected, mean larval speeds before the interaction shows little change over the small window. We do observe a transient of approximately 1–2 min just after the interaction, where there is a noticeable increase in speed ($p < 0.05$, Student's t-test), but it quickly returns to the same mean as before (horizontal line).

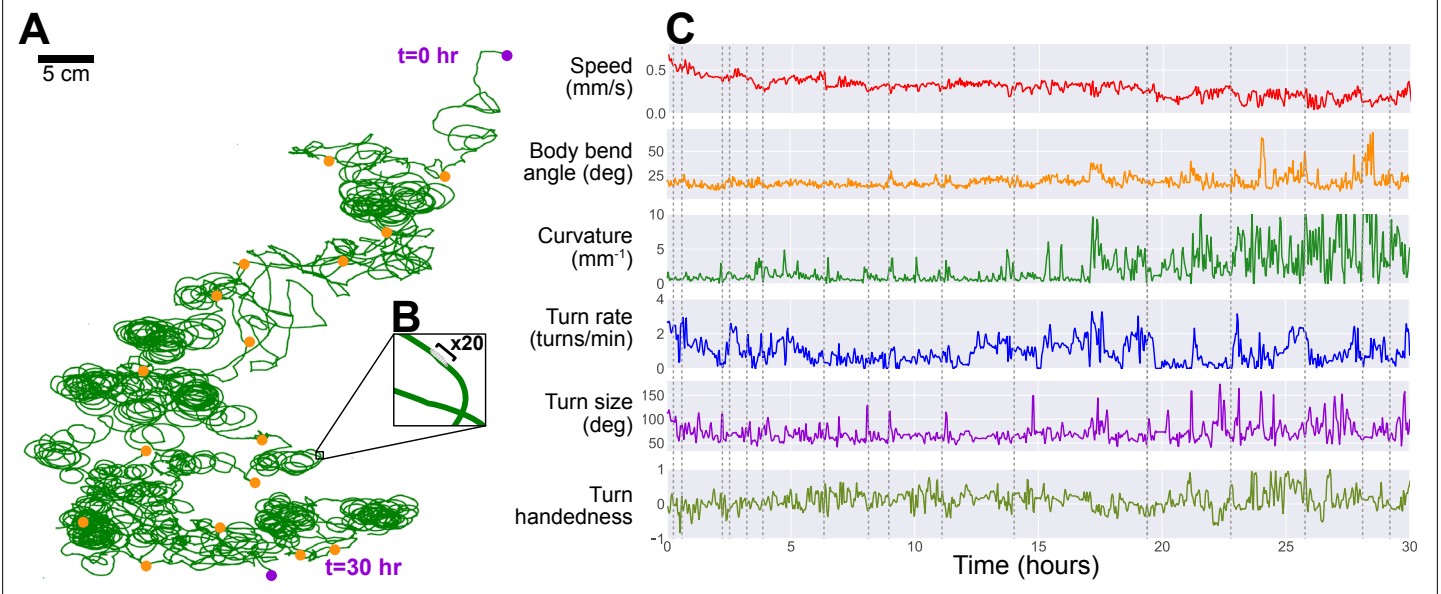

**Figure 4.** Long-term observation of a single larva. In order to maintain exploratory search behavior in a fly larva without starving it, the robot automatically delivers a drop of apple juice (≈0.1 g/mL sugar concentration). The larva is allowed to eat for 1–2 min, after which the robot uses water and air to rinse the larva, which then continues roaming freely. This protocol allows for continuous observation of larval behavior over developmental time scales. (**A**) The larva's trajectory over a 30 hr duration, with its path (green) stitched together by matching each corresponding robot pick-up and drop-off positions (orange markers) by translation (no rotation, e.g. up is always toward same edge of agar) to produce a continuous trajectory. The larva begins at the top right and ends its run at the bottom (purple markers). (**B**) A ×20 magnification on a small section of the path, showing the scale of the path compared to the larva's body (black bar, ≈ 1 mm). (**C**) We plot a number of behavioral features observed during its search trajectory, including its speed (red), body bend angle (orange), trajectory curvature (green), turn rate (blue), turn size (purple), and turn handedness (olive). Turn handedness is calculated as $(N_{left} - N_{right})/N_{total}$, such that a handedness of 1.0 indicates all left turns, and a handedness of −1.0 indicates all right turns. Dotted gray lines indicate time of robot pick-up events.

## A freely crawling single larva is continuously monitored for more than 30 hr

As noted in the previous section, the robot is also capable of automatic scheduled feeding. To keep the larva alive as long as possible, the robot administers a drop of sugar-rich solution directly to the larva once every hour. The larva is allowed to feed for 1 min, then the animal and the drop site is rinsed with water, and the larva resumes its exploratory search. Using this protocol, we are able to study larval locomotive behavior for over 30 hr, yielding an unprecedented amount of behavioral and developmental information on an individual animal. *Figure 4* presents the trajectory of a single larva and some of its behavior features observed over a duration of 30 hr, where the individual animal crawls for more than 48 m. Some behavioral features exhibit steady change, like a decreasing speed throughout the experiment. Notable differences in path shape occur approximately half way through the long observation, with dramatic increases in curvature and turn size, consistent with the tight loops seen in the full trajectory plot (*Figure 4A*). It is unclear what caused the change in behavior. In addition to biological triggers, possible causes include changes in the environment, such as changes in the ambient humidity and uneven changes in agar desiccation or temperature profile producing subtle gradients over the experiment duration. We also note that this larva maintains a left-turning bias throughout the experiment.

## Larval thermotaxis is maintained over long periods

By leveraging the automated transport system's flexible design to deliver and study responses to stimuli, we manipulate the temperature of the agar arena to study larval thermotaxis behavior. *Drosophila* larvae have robust, highly sensitive, and well-documented response to changes in temperature, and work has been done to decipher the behavioral strategy utilized to efficiently navigate thermal gradients (*Luo et al., 2010*, *Klein et al., 2015*). However, there is a lack of abundant data on individual animals, since single experiments last on the order of 10 min, and thus a lack of

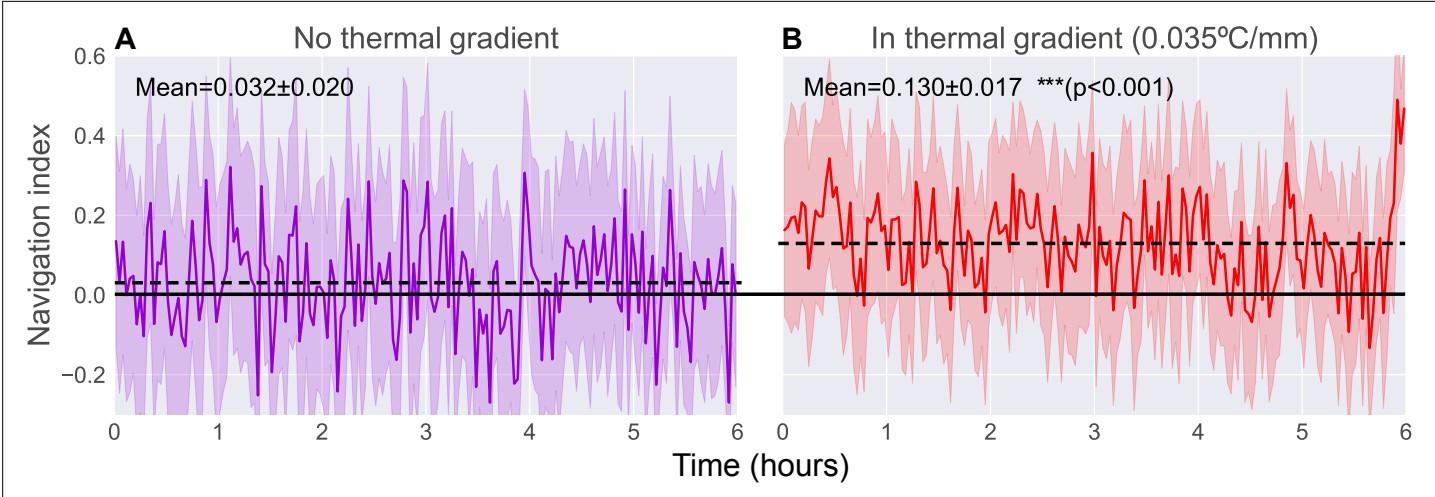

**Figure 5.** Comparison of navigation index in a zero gradient environment (**A**, $N = 42$) and in a presence of a linear thermal gradient of 0.035°C/mm (**B**, $N = 38$). Navigation efficiency is calculated as a dimensionless index equal to $\langle v_x \rangle / \langle v \rangle$, such that +1.0 is parallel to gradient, 0.0 is normal to gradient, and −1.0 is anti-parallel to gradient. We observe a clear increase in average navigation index (dashed lines) when exposed to a thermal gradient (increase from $0.032 \pm 0.020$ to $0.130 \pm 0.017$ [$p < 0.001$, Student's t-test]), but we do not observe any significant pattern of change in that index over time. Solid line indicates a navigation index of zero, indicative of no preference in crawling direction. Shaded region indicates one standard deviation.

understanding of the differences in thermotaxis strategy and its development between individuals. It is also unknown how thermotaxis might evolve over long times. The automated transport system here provides an opportunity to delve deeper into these questions.

Navigational efficiency can be captured by a dimensionless navigation index equal to $\langle v_x \rangle / \langle v \rangle$, the average of the component of crawling velocity along the thermal gradient normalized to the average speed. The resulting navigation index ranges from +1.o (parallel to gradient, i.e. crawling directly toward the warm side of the arena) to −1.0 (anti-parallel to gradient, i.e. crawling directly toward the cold side of the arena) (*Klein et al., 2015*, *Luo et al., 2010*, *Gallio et al., 2011*). *Figure 5* shows the navigation index of second instar wild type (Canton-S) larvae as they crawl across the experiment arena for 6 hr, both with ($N = 38$) and without ($N = 42$) a thermal gradient present. The thermal gradient is centered at 17°C with a steepness of 0.032°C/mm, which would normally evoke robust cold avoidance behavior (*Luo et al., 2010*).

When roaming freely with no gradient present, we observe an average navigation index of $0.032 \pm 0.020$ (range indicating standard deviation). When a thermal gradient is applied to the arena, we observe a clear increase in navigation index to an average of $0.130 \pm 0.017$ (p < 0.001, Student's t-test). The navigation index also remains steady over the 6 hr measurement time.

## Navigation efficiency of individuals exhibits distinct behavioral phenotypes

Short snapshot observations produce limited information on any single animal, therefore limiting most thermotaxis analysis to population-level statistics. Long continuous observation enabled by the transport robot uncovers much more detailed information about individual animals, allowing us to analyze behavioral features at the individual level as well (*Figure 6*). Importantly, averaging over a population necessarily results in some loss of information such that different statistics at the individual level can generate the same population mean.

*Figure 6A–B* demonstrates the differences between two such cases by examining simulated toy examples of a probability distribution of observations of a navigation index, where a point on the distribution represents the probability of observing a certain navigation index at any given time. Each series of observations (one series of observation represents behavior of a single individual) is sampled from a Gaussian distribution, whose mean and standard deviation are randomly determined with two different statistics. In the first simulation (*Figure 6A*), the Gaussian mean and standard deviations for each animal have low inter-animal variability, producing a mean of intra-animal distributions that is similar to the population mean distribution. In the second (*Figure 6B*), high inter-animal variability

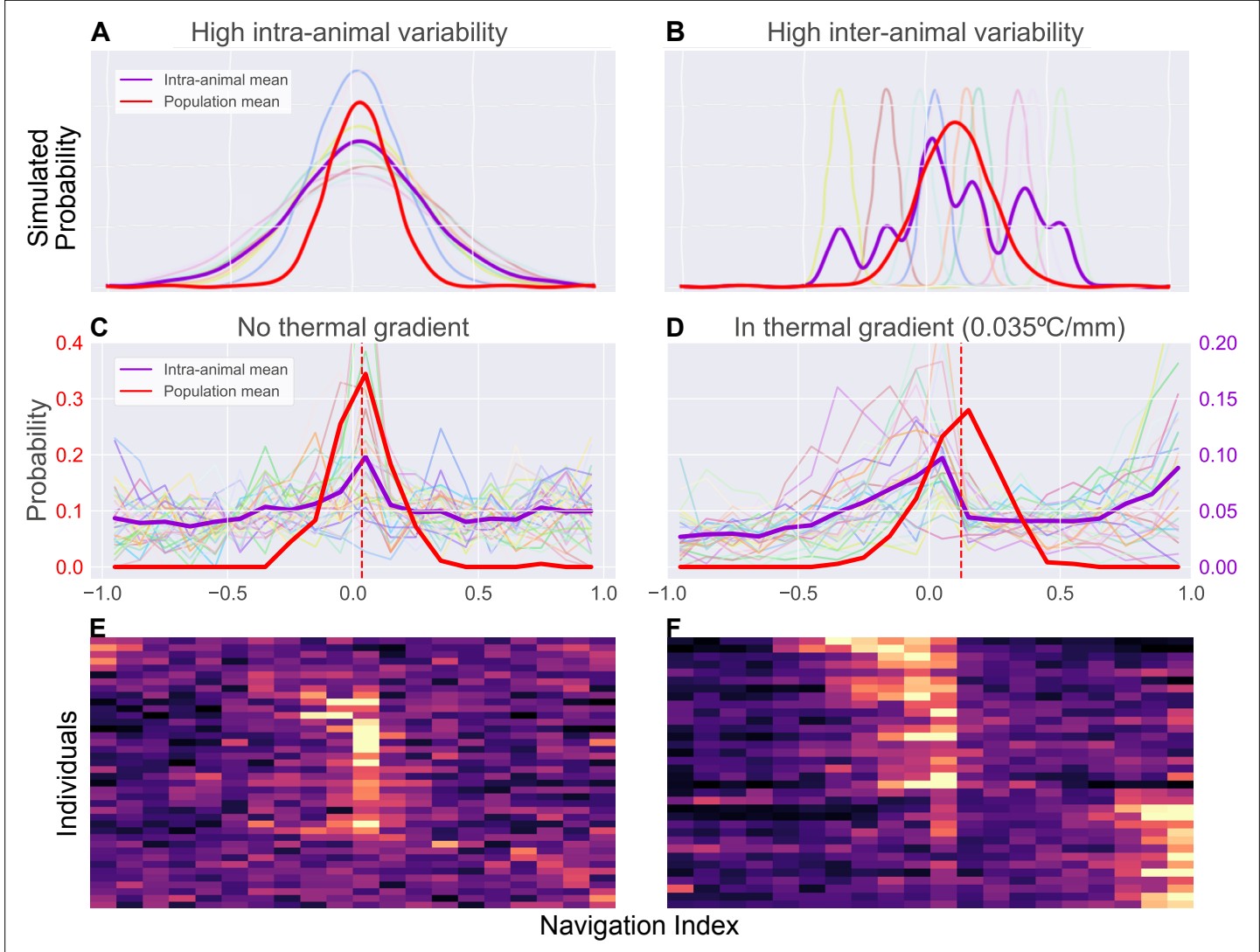

**Figure 6.** Examination of inter- and intra-animal variability via analysis of the probability distribution of observed larval thermal navigation index. Each distribution shows the probability (vertical axis) of observing a certain navigation index (horizontal axis) at any given time in a series of observations for single individuals (purple), or a series of average observations for the population at each time point (red). When dissecting probability distributions of observed behavior, we notice that the same population (inter-animal) mean can be produced by two individual (intra-animal) distributions. (**A**) Simulated example of individual probability distributions with high intra-animal variability. The thin, lighter traces are probability distributions of eight individual animals. The resulting mean of intra-animal distribution (thick purple) closely resembles the population mean (thick red). (**B**) Simulated example of individual probability distributions with high inter-animal variability. The thin, lighter traces are probability distributions of eight individual animals. The resulting mean of intra-animal distribution (purple) forms a multimodal distribution despite a similar population mean (red) as in (**A, C**). Empirical probability distribution of navigation index observed without a thermal gradient. There is high intra-animal variability but low inter-animal variability, such that the intra-animal mean forms a similar distribution to the population mean, as in the simulated results in panel A. $N = 42$ individual larvae. (**D**) Empirical probability distribution of navigation index observed in presence of a thermal gradient (0.035°C/mm). In contrast to (**C**), during thermotaxis, the intra-animal mean forms a bimodal distribution ($BC = 0.67$, compared to $BC = 0.48$ in **C**) despite each individual distribution remaining unimodal ($BC = 0.51 \pm 0.06$ with gradient, $BC = 0.50 \pm 0.04$ without). $N = 38$ individual larvae. This more closely resembles a distribution with high inter-animal variability as in the simulated results in panel B. (**E**) Individual empirical probability distribution of navigation index observed without a thermal gradient, displaying the same data used to generate panel C. (**F**) Individual empirical probability distribution of navigation index observed in presence of a thermal gradient, displaying the same data used to generate (**D**).

in the same instead produces a multi-modal distribution for individuals despite the extra modes not being present at the population level. While the two simulations show distinct means of intra-animal probability distributions (purple), that is variability between individuals is different, both simulations still produce similar distributions when analyzing probability of navigation index observations at the

population level (red). Because our long timescale thermotaxis experiments generate enough data to establish both population-level and individual-level navigation indices, we can examine how similar distributions of larval thermal navigation behavior may compare between the population and individual levels.

Interestingly, we find that larval navigation index behavior switches from one model to the other when a thermal gradient is applied (*Figure 6C and D*). The population means have a similar shape and spread in both gradient and gradient-free contexts (red traces in *Figure 6C and D*). However, without a thermal gradient (*Figure 6C*), the individual distributions exhibit high intra-animal variability and low inter-animal variability, mirroring those seen in the first simulation (*Figure 6A*). In the presence of a thermal gradient (*Figure 6D*), we observe a bimodal intra-animal distribution despite each individual distribution remaining unimodal ($BC = 0.51 \pm 0.06$ with gradient, $BC = 0.50 \pm 0.04$ without), more closely resembling the second simulation (*Figure 6B*) instead. We measure a binomial coefficient (BC) (*Freeman and Dale, 2013*, *Codling et al., 2008*) of 0.67 in a thermal gradient, compared to $BC = 0.48$ when there is no gradient present. This is a significant increase (p < 0.01, Student's t-test) that crosses the critical value for detecting bimodality ($BC_{crit} = 5/9$), clearly indicating a shift in the shape of the distribution.

## Discussion

We have developed an automated system for long-term observation of *Drosophila* larvae (*Figure 1*). The robotic arm is capable of transporting larvae as they approach the edge of the experiment arena back to the center, allowing continuous observation of exploratory search or directed navigation behavior from an overhead camera. Through coordination and constant feedback between the robot and video acquisition, the system maintains larvae within the arena with high reliability, and we are able to achieve continuous observation over developmental time scales. The accompanying analysis pipeline takes the output video from these experiments to track larval posture and behavioral state while maintaining individual identities. The analysis compensates for the output video's low resolution and lack of detailed features of the larvae through a combination of local and global features and the use of recurrent neural networks (*Figure 2*).

We present a study of free-roaming behavior in larvae over 6 hr of continuous observation, comparing results to short 10 min snapshot observations of larvae at various stages of starvation (*Figure 3*). The comparison yields dissimilar patterns when considering the 10 min averages, but similar dynamics when considering the change in behavior over time, such that the 10 min trajectory in larval crawl speed resembles that seen in the first hour of continuous observation. The similarity suggests behavioral dynamics that are present in both experiments, but persist and develop over a duration that is an order of magnitude longer than what snapshot observations can capture.

Since our analysis maintains animal identities throughout the video, we are able to capture behavioral information on single individuals with unprecedented detail. We leverage this new trove of data to analyze larval response to a thermal gradient and examine the probability distribution of the observed navigation index over time (*Figure 6*). In particular, we note that the same population (inter-animal) mean can be produced by different individual (intra-animal) distributions. Interestingly, larvae seem to switch to a different distribution shape upon encountering a thermal gradient. The individual distributions of navigation index without a thermal gradient exhibits high intra-animal variability but low inter-animal variability, forming a unimodal mean distribution that is similar to the population mean. In contrast, during thermotaxis the individual distributions exhibit high inter-animal variability, such that it become bimodal when a gradient is applied, despite the distribution shape remaining unimodal at the individual and population level. Thermotaxis behavior of individual crawlers approximately falls into two categories of neutral and strong crawlers (bimodal distribution), but this phenomenon was masked in past research because analysis focused on population averages and the experiments were too short. With our individual-focused and long-term experiments, we were able to see this phenomenon more clearly. This result is consistent with recent findings that suggest a switch-like (all-or-none) learning behavior in larval *Drosophila*, which is also not apparent when only analyzing population means. By observing decision-making behavior of individual larvae over several cycles of stimulus and reward, *Lesar et al., 2021*, find that Pavlovian training of preference for carbon dioxide is similarly quantized to two states (all-or-none), each centered at a fixed preference index. While the context and modality for the learning assays are different than our observation of larval thermotaxis, both

results reveal new features of larval behavior that were previously obscured in population averages. Both results also suggest the existence of larval 'personality types', or distinct, quantized behavioral phenotypes that vary between individual animals (*Lesar et al., 2021*, *Klein et al., 2015*, *Günther et al., 2016*, *Ohyama et al., 2013*). Our present work provides some progress in uncovering such phenotypes and furthering our understanding of learning and development of navigational strategies in individual animals in addition to population averages.

With the robot's flexible design, we can continue to probe these questions in many different contexts. For example, the robot could deliver food or soluble drugs directly to the larvae on a predetermined schedule to measure both the acute and chronic effects on the animal's behavior and physiology (*Wolf and Heberlein, 2003*, *Bainton et al., 2000*, *Besson and Martin, 2005*, *Gerber and Stocker, 2007*, *Fayyazuddin et al., 2006*, *Robinson et al., 2012*, *Kaun et al., 2012*). We could also leverage the existing lighting system to implement optogenetic activation or suppression of specific neurons, allowing delivery of fictive stimuli or studies of the effects of certain neuronal circuits on larval behavior and development (*Gradinaru et al., 2010*, *Inada et al., 2011*, *Salvaterra and Kitamoto, 2001*, *Hernandez-Nunez et al., 2015*, *Kane et al., 2013*). This could either be activated on a predetermined schedule, or integrated directly with the existing robot and camera feedback system to enable activation triggers based on specific conditions such as larval behavior (e.g. activate upon larva initiating a turn).

Our study has sought to characterize very-long-timescale free-crawling behavior, both with and without a single stimulus that influences crawling direction. The larva picker robot effectively acts as a 2D conveyor belt, rapidly (on the order of seconds) repositioning a larva each time it reaches the edge of the arena (on the order of every 60 min). Although this method enables continuous measurement of free crawling, it does have inherent limitations.

First, the behavior we measure is far removed from what larva behavior looks like in an ecological setting. In nature, larvae search for rotting fruit to consume, and if they find it, tend to stay in it during larval development, then typically move to drier areas to pupate (*Sokolowski et al., 1986*). During this process they sense and respond to many stimuli (odors, humidity, light, vibration, temperature, etc.). In the present work we isolate the larvae from stimuli, placing them on a flat substrate with no visible light, food, or temperature changes, thus providing an environment where larvae are always searching, in a state of near-constant locomotion. This could affect larval development times and other behavioral features, which will be in a better position to study with a more rigorous feeding system and a larger data set. As an early fundamental step in what we hope becomes a more complete understanding of multisensory integration and behavior in realistic settings, we have focused on understanding high-bandwidth 2D crawling trajectories and the basic behavior sequences that form them.

Second, there are limitations to how well we can truly isolate crawling larvae from outside stimuli. Although experiments are run with red lights (not visible to larvae), other light from the surrounding lab could reach the animals, and the room is not precisely temperature stabilized. Similarly, vibrations or air currents in the building were not measured or directly controlled. The most blatant outside stimulus is the pick-up and drop-off process from the robot. Although we have determined that the short-term effects of this disturbance fade after less than 60 s, we do not yet have sufficient data to study long-term effects, and repeated transport from the robot arm could introduce important changes. We do have several reasons to believe these effects will be minimal, however. In one of our previous studies (*Klein et al., 2015*) we observed that a larva's thermal response behavior becomes uncorrelated with its stimulus history after approximately 10 s; and in another *Berne et al., 2021* found that physical behavior returns to normal within approximately 20 s of a stimulus introduction, even in presence of very strong vertical vibration. Further, although fly larvae can form associative memories, particularly in pairing conditioned stimuli with odors or tastes (*Gerber and Stocker, 2007*), our arenas are free of salient stimuli that would be likely to induce this such memory formation. Since larvae do not see in our environment and do not make spatial maps, we consider the larva's behavior to be exploratory, even though in a literal sense they are exploring the same space repeatedly. Because the animal is highly food-motivated, it will continuously execute search patterns even after many hours of failure to locate food.

A major obstacle in way of achieving continuous observation of the entire larval life cycle ($\approx$ 100 hr) is a reliable method of delivering enough nutrition and ensuring the larva has sufficiently fed. The current method of delivering drops of sugar-rich solutions has not been sufficient to trigger molting

into the next instar stage, thus limiting us to a single instar. Feeding larvae on their primary behavior arena can also introduce new stimuli, as remnants of the food solutions can be sensed by the olfactory or gustatory systems in the larvae. Future improvements to the apparatus may include a separate short-term feeding station away from the main navigation area. We hope to solve this problem and others as we continue to develop the system and expand its capabilities, for example with a second adjacent arena with more nutritive food where larvae could experience longer feeding times before being returned by the robot to the primary behavioral arena. Such rich detail covering the entire development of the larva would provide powerful insight into the long-term learning, memory, and behavioral adaptations in tandem with the physiological developments that take place between instar stages.

We hope that this study has highlighted some advantages of long-term continuous measurement, and that similar instrumentation and analysis method could be applied to other organisms, along with a wide range of investigations in the fly larva.

## Materials and methods

### Larva picker robot

The manipulator arm is translated by stepper motors (Nema 8) in the X- and Y-axes and a 5V solenoid (SparkFun) for the Z-axis, which are driven by a programmable controller board (SmoothieBoard) with physical and software limits in place to prevent overtravel. At the end of the arm is a custom 18-gauge nozzle (*Figure 1C*) that can interact with the larva and the experimental arena (*Figure 1D*) that sit below the camera and manipulator assembly.

A 2.3-megapixel CMOS camera (Grasshopper3) observes a small number (4–6) of larvae crawling on a 22 × 22 cm$^2$ agar gel (2.5% wt/vol agar in water, with 0.75% charcoal added for improved contrast) and records at 10 Hz. Larvae are illuminated with four strips of red LEDs (dominant wavelength around 620 nm), which is outside the visible range of the larva (*Sprecher et al., 2007*), arranged in a square around the agar gel. To maintain larval exploratory searching behavior over a long duration, the camera detects when one nears the edge of the arena. This triggers the manipulator to pick up the larva with the nozzle. The larva is maintained on the nozzle via the surface tension of a water droplet. The water droplet provides a way to indirectly interact with the larva to prevent causing damage to the animal. After the manipulator arm moves to the center of the arena, it drops off the larva with a slow horizontal motion, effectively rolling it off the nozzle (*Figure 1E*). The manipulator replenishes the water droplet before each pick-up, and a small flat Delrin plastic disk (2 mm diameter) at the bottom of the nozzle provides more surface area for the droplet to form (*Figure 1C*). When the surface tension of the water is insufficient to pick up the larva, the nozzle is capable of exerting vacuum suction to assist in pick-up, as well as allowing air flow to release the larva during drop-off.

In some experiments we observe how larvae navigate a variable sensory environment. We use a similar system as outlined in *Klein et al., 2015*, to generate a 1D linear spatial thermal gradient.

We fit the underside of the experimental arena's aluminium base with hot and cold reservoirs on opposite sides, each equipped with two liquid-cooled water blocks. PID (proportional-integral-derivative) controllers drive thermoelectric coolers between each water block and its reservoir to maintain a thermal gradient of 0.035°C/mm across the agar gel in the arena, with 13°C on one side and 21°C on the other.

### Constrained larval behavior

*Drosophila* larvae tend to spend a lot of time near the edges of conventional cultivation dishes. To demonstrate this, we monitored larvae crawling in a 60 mm agar dish for 20 min. An example time lapse showing the distribution of animals moving toward, and dwelling at, the perimeter is shown in *Figure 1—figure supplement 1*.

### Analysis pipeline
#### Position tracking
Custom computer vision software takes raw movies and extracts the position and body contour of each larva while preserving individual animal identities.

Each frame is prepared with increased contrast with dynamic background subtraction. Over long duration experiments, the background that the larvae navigate may change slowly over time due to interactions with the animals or environmental effects such as condensation or dust. With this in mind, we update a dynamic background that will be subtracted from each frame using an infinite impulse response filter. The background, $B_t$, for the frame, $F_t$, is calculated as:

$$B_t = \alpha F_t + (1 - \alpha)B_{t-1}$$

where $\alpha$ defines the feedforward coefficient that is chosen heuristically based on the imaging rate and experimental conditions.

After preprocessing, larvae in each frame are detected by isolating outermost contours around clusters of bright pixels. Contours within a size range are identified as individual larvae, where the size range is manually adjusted to match the camera resolution and the current larval development stage. Animal identities are linked frame-to-frame by minimizing a pairwise loss that is calculated using a weighted sum of distances between heuristically chosen features from the previous frame to the next. The loss between a detected contour, $i$, and a larva from the preceding frame, $j$, is calculated as:

$$L_{ij} = \beta_r D(r_i, r_j) + \beta_p D(p_i, p_j) + \beta_a D(a_i, a_j)$$

where $r$ is position of the larva's center of mass, $p$ is momentum, $a$ is the contour area (i.e. animal size), and $D(x, y)$ is the Euclidean distance between vectors $x$ and $y$. The coefficients, $\beta$, weigh each distance term and can be adjusted for different animals and imaging conditions. Each new contour, $i$, is uniquely assigned to an ID, $j$, beginning with the pair with the lowest loss, then the next lowest loss out of the remaining possible pairs, etc. Any larvae without an assignment in the current frame is given interpolated properties based on the previous few frames to compensate for occlusions or other cases when it is missing from view. Since the linking is sequential, doing so also helps avoid some cases of undesirable identity swaps. With contours in each frame assigned to a unique larva, the centroid of each contour is recorded as the position of the animal and is used to produce a crop around each larva.

We observe collision events occurring at an average rate of 0.63 events per hour in an arena of six larvae. In frames where two or more animals are colliding, that is their contours are on top of each other to form a single large contour, the contours for the colliding animals from previous frames are translated and combined to generate a similarly large contour. The translation of each contour is initialized using the momentum of the larva in preceding frames, and is optimized for a small number of epochs ($N < 50$) such that the resulting combination best matches the large contour seen in the current frame. The coordinates resulting from this optimization are used as the position of the animal in these frames instead. Through this method, identities are preserved across collision events with an accuracy of 91%, or <1 misidentifications per 6 hr experiment, which can be manually verified and corrected.

## Posture analysis

The isolated crops (64 × 64 in pixels) around each animal are run through a recurrent U-Net (*Ronneberger et al., 2015*) convolutional neural network to determine the posture and orientation. The U-Net architecture has been shown to be highly effective at tasks that preserve spatial structure in an image by taking advantage of both global and local features (*Ronneberger et al., 2015*, *Risse et al., 2017*, *He et al., 2015*). We design our model such that the output is a probability heat map of the head and tail of the larva, which preserves the global spatial structure of the larva's body.

Traditional convolutional neural networks, including U-Net, often fail when analyzing sparse images with low-resolution features (*Risse et al., 2017*). This is particularly pronounced in our case since each larva is generally captured as clusters of only ≈30 uniformly bright pixels surrounded by black pixels. To compensate for this, we utilize more temporal information, such as the current momentum of the centroid. Since larval posture does not deviate much from frame to frame, we add recurrence in the form of long short-term memory (LSTM) cells (*Hochreiter and Schmidhuber, 1997*) at the beginning, middle, and final layers of the network to simplify the problem at each subsequent time step. The recurrence also creates an additional cost for head-tail flips which we have found to be a common issue with previous approaches to the problem (*Risse et al., 2017*, *Mathis et al., 2018*, *Günel et al., 2019*).

Our implementation of the network comprises two main modules: an encoder and a decoder. The encoder is composed of four submodules of alternating convolutional neural network layers and nonlinear activation layers, including a rectified linear unit (ReLU) and a pooling layer. This is mirrored by the decoder on the other side with the same submodules with upsampling layers replacing the pooling ones. We follow the U-Net architecture to add information 'highways' between corresponding layers on the encoder and decoder side, passing outputs from each submodule of the encoder into the corresponding submodule of the decoder. We further modify this architecture by adding recurrence in the form of LSTM cells in the convolutional layers at the beginning of the encoder, the centermost layers between the two modules, and the end of the decoder to simplify the problem at each subsequent time step.

We formulate the output as a probability heat map of keypoints that define the posture of the animal (e.g. head and tail of the *Drosophila* larva), which preserves the global spatial structure of the larva's body. The network is trained using manually labeled videos of larvae with ≈1000 frames per animal undergoing several posture changes, including at least one turning event. Upon verification with a small number of such videos withheld during training, we measure of RMSE of $1.60 \pm 0.66$ px for the head and tail coordinates.

We may also estimate the 'spine' of the animal that characterizes the curvature of the body by using the head and tail points as shown in *Risse et al., 2017*. The contour of the animal is split into two halves, $c_1$ and $c_2$, such that both connect the head and tail along either side of the body. We resample the half-contours at equal rates via linear interpolation such that each point along $c_1$ has a unique match along $c_2$. Spine points can then be calculated as the midpoint between equidistant pairs along the two contours. We use the resulting 'midspine' point as a more accurate position of the animal, $r_t$, which is favored for subsequent downstream analysis of behavior over the centroid of the animal contour in image space.

## Behavior classification

Using all available information (position, contour, and posture), we use a densely connected neural network with bidirectional recurrence to classify the behavioral state of the animal (e.g. 'forward crawl', 'turn') at each time point (*Günther et al., 2016*). The network is a sequence of nine densely connected linear layers alternating with ReLU activation layers, and a bidirectional LSTM layer at the beginning and middle of the sequence for a total of 11 layers. The bidirectional recurrence here allows the network to identify the bends in the animal's path and particular postural dynamics by comparing frames both before and after. A softmax activation layer at the end of the network scales the output between 0 and 1, representing the probability of each possible larval behavior exhibited in the current frame. The state with the highest assigned probability is selected as the current behavioral state, with some additional cost for switching between behavioral states. From here, we can identify a wide range of behavioral features and track them over extended time periods.

The network is trained using manually labeled videos of individual larvae crawling for 1 hr. Since the larva is likely to be crawling forward for a large majority of frames, the loss is weighted 10-fold toward frames during turning events. Upon verification with a small number of such videos withheld during training, we measure of an accuracy of 99.8% of frames labeled with the correct behavioral state.

## Acknowledgements

VV acknowledges support from the Burroughs Wellcome Fund and NIH R01 NS126334. MK acknowledges support from the NSF CAREER Award 2144385.

# Additional information

## Competing interests

Dave Zucker: Dave Zucker is affiliated with FlySorter, LLC. The author has no financial interests to declare. The other authors declare that no competing interests exist.

## Funding

| Funder | Grant reference number | Author |
| --- | --- | --- |
| Burroughs Wellcome Fund | | Vivek Venkatachalam |
| National Institutes of Health | NIH R01 NS126334 | Vivek Venkatachalam |
| National Science Foundation | NSF CAREER 2144385 | Mason Klein |

The funders had no role in study design, data collection and interpretation, or the decision to submit the work for publication.

## Author contributions

James Yu, Conceptualization, Data curation, Software, Formal analysis, Investigation, Visualization, Methodology, Writing - original draft, Writing - review and editing; Stephanie Dancausse, Formal analysis, Validation, Investigation, Visualization, Methodology, Writing - review and editing; Maria Paz, Investigation, Methodology; Tolu Faderin, Melissa Gaviria, Investigation; Joseph W Shomar, Resources, Methodology; Dave Zucker, Conceptualization, Resources, Software; Vivek Venkatachalam, Mason Klein, Conceptualization, Resources, Supervision, Funding acquisition, Investigation, Methodology, Writing - original draft, Project administration, Writing - review and editing

## Author ORCIDs

James Yu ⓘ http://orcid.org/0000-0002-6247-0433
Joseph W Shomar ⓘ http://orcid.org/0009-0000-9534-6263
Dave Zucker ⓘ https://orcid.org/0000-0003-0259-6293
Vivek Venkatachalam ⓘ http://orcid.org/0000-0002-2414-7416
Mason Klein ⓘ http://orcid.org/0000-0001-8211-077X

## Decision letter and Author response

Decision letter https://doi.org/10.7554/eLife.86585.sa1
Author response https://doi.org/10.7554/eLife.86585.sa2

# Additional files

## Supplementary files

• MDAR checklist

## Data availability

Software drivers and analysis tools are hosted in the repository https://github.com/venkatachalamlab/LarvaPickerRobot (copy archived at *Yu, 2023*). Mildly compressed versions of all video recordings can be found hosted on Zenodo.

The following dataset was generated:

| Author(s) | Year | Dataset title | Dataset URL | Database and Identifier |
| --- | --- | --- | --- | --- |
| Yu J, Dancausse S, Paz M, Faderin T, Gaviria M, Shomar JW, Zucker D, Venkatachalam V, Klein M | 2023 | Videos of long-term *Drosophila melanogaster* crawling behavior | https://doi.org/10.5281/zenodo.8208884 | Zenodo, 10.5281/zenodo.8208884 |

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
