## [Editor Report]

This study describes a useful method to monitor the behavior of *Drosophila* larvae in a uniform environment over much longer time scales than was possible with previous methods. The authors provide a solid characterization of aspects of the method and show that the behavior of single larvae can be quantified over several hours. The experiments offer a proof-of-concept for a robotic device that will enable the investigation of behavior in long-term experiments in ways that were previously unimaginable.

---

## [Decision Letter]

**Decision letter after peer review:**

Thank you for submitting your article "Continuous, long-term crawling behavior characterized by a robotic transport system" for consideration by *eLife*. Your article has been reviewed by 3 peer reviewers, and the evaluation has been overseen by a Reviewing Editor and K VijayRaghavan as the Senior Editor.

Essential revisions:

The reviewers appreciate the qualities and future impact of the method you have developed to study the long-term behavior of *Drosophila* larvae. The observation of a bimodal distribution in the navigation index of single larvae highlights that mean behaviors measured over short observation times can mask diversity in the behaviors of individual animals. You provide strong evidence that your system will be useful to other laboratories interested in prolonged recording of larval activity. However, several weaknesses of the manuscript should be addressed prior to its publication. The revision does not require the collection of additional data. Generally, you should provide a more thorough description of where your method fits with respect to published work and how the method itself works. More specifically, we recommend that you implement the following changes:

– Improve the introduction to more accurately reflect previous work done to study behavior in worms and rodents over long durations.

– Provide additional information about the training process of the neural network used to track larvae. Please ensure that the method is reproducible and customizable by other labs.

– Discuss the implications of physical perturbations during the picking and transportation of an animal. These perturbations are likely to affect subsequent behavior, which might preclude the study of prolonged free-moving exploration. These limitations should be discussed in the manuscript.

– Comment on the ability of the tracker to maintain the identity of single larvae over prolonged periods of time.

*Reviewer #1 (Recommendations for the authors):*

Introduction:

43: More focus is needed for clarification. What is the problem you are trying to solve? It is very much larva-specific but that is not clear at all from the introduction. Both the aim of the work and the relative background in the literature can be defined in greater detail. High throughput analysis of behavior, in general, is possible and not prohibitive at all. Examples are the *Drosophila* activity monitors (commercial) and the ethoscope (open source). Both are able to record the activity of adult animals throughout their lifespan, in high throughput.

87: There is an issue here with how you define exploration. Picking an animal from the border of an arena to place it back in the center is not a way to prolong exploration. It's certainly a way to prolong crawling behaviour but not necessarily exploration. You are basically mimicking a conveyor belt here: it can certainly be useful but I don't think it has much to do with exploration. In fact, it would be important to think of this concept in ecological terms and ask: for how long will a larva crawl/explore its natural environment? Will a larva in an apple crawl continuously for 30 hours, as it does in this device?

315: I suppose you refer to *melanogaster* only, or do you mean all *Drosophila* species?

Figure 1: I don't think the concept exposed in E is so difficult to require illustrations. F is really difficult to understand and thus not informative. Perhaps replace it with a supplementary video of the picking process?

Figure 5: Not clear how to interpret this figure. What are the dashed and continuous lines? What does the P*** refer to? You refer to a "clear increase in navigation index" but this is not clear by looking at the figure.

Figure 6: I am afraid I do not understand the reasoning behind this experiment. This would really benefit from a clearer and more accessible explanation, in terms of aims and methods.

*Reviewer #2 (Recommendations for the authors):*

My only comment to the authors is that it would nice to see additional paths in the very long duration recordings. Are some larvae right-handed while others are predominantly left-handed?

*Reviewer #3 (Recommendations for the authors):*

– I don't think Figure 1A is cited in the text.

– Line 185: How long are the occlusions by the robot arm and how far does a typical larva move (perhaps as a fraction of body length) during the occlusions?

– Line 326: Is "navigational effectiveness" a standard term? If not, I wonder if a better term might be something like navigational cosine? The reason I ask is that if a larva has reached a desired temperature in a gradient then the "effective" thing to do might be to stop moving or move orthogonally to the gradient which would have a Navigational Effectiveness of 0.

– Line 343: include details of statistical test.

Suggestions for improvement:

– Include lines on the time series in Figure 4 to show the picking times.

– In general, the data in Figure 4 is tantalizing, but because it's based on a single observation we can't rule out the increase in speed later in the recording isn't due to chance or some environmental change in the room. The most interesting thing would be to include some more data, but if that's not possible, including some more discussion of possible causes (including those that don't have anything to do with larval biology) in the text might help.

---

## [Author Response]

Essential revisions:The reviewers appreciate the qualities and future impact of the method you have developed to study the long-term behavior of *Drosophila* larvae. The observation of a bimodal distribution in the navigation index of single larvae highlights that mean behaviors measured over short observation times can mask diversity in the behaviors of individual animals. You provide strong evidence that your system will be useful to other laboratories interested in prolonged recording of larval activity. However, several weaknesses of the manuscript should be addressed prior to its publication. The revision does not require the collection of additional data. Generally, you should provide a more thorough description of where your method fits with respect to published work and how the method itself works. More specifically, we recommend that you implement the following changes:– Improve the introduction to more accurately reflect previous work done to study behavior in worms and rodents over long durations.

The Introduction section has been significantly reworked, to better explain how our work fits into the context of studies in other organisms, and highlighting what makes our project unique – both because of the animal we use and because we create an environment of prolonged free exploration, and we make more sound arguments for why this is important. See in particular the second paragraph of the introduction (all new) and a heavily changed fourth paragraph.

– Provide additional information about the training process of the neural network used to track larvae. Please ensure that the method is reproducible and customizable by other labs.

We have added a new section in Materials and methods devoted to our analysis pipeline, providing details for position tracking, posture analysis, and behavioral classification (1-1.5 new pages total). As noted in the previous submission, all code for the manuscript will be provided to any reader, hosted on GitHub.

– Discuss the implications of physical perturbations during the picking and transportation of an animal. These perturbations are likely to affect subsequent behavior, which might preclude the study of prolonged free-moving exploration. These limitations should be discussed in the manuscript.

We have expanded the Discussion section of the manuscript to include a more direct discussion of the limitations imposed by the robot arm pick-ups/drop-offs and external environmental conditions. We enumerate a few reasons why we believe the perturbations’ effects will be minimal, but also openly state that we do not have sufficient information to claim that physical perturbations are not important. Most of the new text is found in the 6^th^ and 7^th^ paragraphs of Discussion.

– Comment on the ability of the tracker to maintain the identity of single larvae over prolonged periods of time.

We now discuss this in detail in the Materials and methods section (last two paragraphs of the Position Tracking sub-section). We describe how identity is maintained, and also provide quantitative measures of collision rates between larvae and the overall accuracy of identity maintenance after collisions.

Reviewer #1 (Recommendations for the authors):Introduction:43: More focus is needed for clarification. What is the problem you are trying to solve? It is very much larva-specific but that is not clear at all from the introduction. Both the aim of the work and the relative background in the literature can be defined in greater detail. High throughput analysis of behavior, in general, is possible and not prohibitive at all. Examples are the *Drosophila* activity monitors (commercial) and the ethoscope (open source). Both are able to record the activity of adult animals throughout their lifespan, in high throughput.

This is absolutely right, the introduction to our work needed to be placed in a much broader context that included other organisms and prior work over long time scales. We have rewritten our Introduction (2^nd^ paragraph in particular), looking at adult flies, worms, mice, and rats. We have also tried to make it much more clear that our primary goal is to (1) measure larva behavior over extended periods, which has its own unique challenges as any animal does, and also (2) measure long term behavior in a non-confined space, to observe what happens to search and navigation strategies on previously unseen time scales.

87: There is an issue here with how you define exploration. Picking an animal from the border of an arena to place it back in the center is not a way to prolong exploration. It's certainly a way to prolong crawling behaviour but not necessarily exploration. You are basically mimicking a conveyor belt here: it can certainly be useful but I don't think it has much to do with exploration. In fact, it would be important to think of this concept in ecological terms and ask: for how long will a larva crawl/explore its natural environment? Will a larva in an apple crawl continuously for 30 hours, as it does in this device?

This is a very good point also. We were loosely conflating searching and exploration. Throughout the manuscript text we have changed every reference to exploration to either “search” or “exploratory search” and clarified the terms better. We also suggested why, for larvae in particular, their motion could be thought of as exploratory even when traversing the same space repeatedly (as far as we know they do not form spatial maps of their environment and their non-associative memories are very short). But we agree the term suggests something rather different and have adjusted accordingly.

315: I suppose you refer to melanogaster only, or do you mean all *Drosophila* species?

We meant *Drosophila melanogaster* only, and have made this explicit at the beginning of the manuscript. Thank you.

Figure 1: I don't think the concept exposed in E is so difficult to require illustrations. F is really difficult to understand and thus not informative. Perhaps replace it with a supplementary video of the picking process?

We do prefer to keep panel E if that is acceptable. We agree about F, which we are leaving in place but with a new supplemental video of the picking process, as you suggest. The video shows the robot arm picking up three larva from the edge of the arena and placing them in the center.

Figure 5: Not clear how to interpret this figure. What are the dashed and continuous lines? What does the P*** refer to? You refer to a "clear increase in navigation index" but this is not clear by looking at the figure.

Thank you for pointing this out. The caption to Figure 5 has been updated to define the dashed and solid lines. The solid line marks 0 (no net navigation), and the dashed lines are the average values of navigation index throughout the 6-hour experiments. The p-value refers to the difference in navigation index between the control (no thermal gradient, purple) and thermotaxis behavior (0.035 C/mm gradient, red).

Figure 6: I am afraid I do not understand the reasoning behind this experiment. This would really benefit from a clearer and more accessible explanation, in terms of aims and methods.

We have rewritten the text of the “Navigation efficiency of individuals exhibits distinct behavioral phenotypes” subsection in the Results section, and also the caption of Figure 6 for clarity, in addition to expanded writing in the Discussion section. The short version is that the thermotaxis behavior of individual crawlers approximately falls into two categories of neutral and strong crawlers (bimodal distribution), but this phenomenon was masked in past research because analysis focused on population averages and the experiments were too short. With our individual-focused and long-term experiments we were able to see this phenomenon more clearly.

Reviewer #2 (Recommendations for the authors):My only comment to the authors is that it would nice to see additional paths in the very long duration recordings. Are some larvae right-handed while others are predominantly left-handed?

Additional videos with the 30+ hour time frame are not doable at present, although we are making changes to the system to allow for many more of these in the future. The handedness question is a really good one – the short answer is yes, individual animals exhibit handedness, in the sense that the distribution of handedness is wider than the binomial distribution for the sample size (this is also true in previously published work in adult flies). But complications (persistence of handedness, handedness during crawling vs. during turns) make the question a bit beyond the scope of this manuscript. We do plan to write a new paper entirely about handedness though!

Reviewer #3 (Recommendations for the authors):– I don't think Figure 1A is cited in the text.

Fixed, thank you. It is now cited in the 4^th^ paragraph of the Introduction.

– Line 185: How long are the occlusions by the robot arm and how far does a typical larva move (perhaps as a fraction of body length) during the occlusions?

This information is now provided in the Analysis Pipeline subsection of Materials and methods. It’s around 4.5 seconds of occlusion, which translates to a sufficiently small fraction of body length traveled, such that interpolating the crawling trajectories during dropped frames works well.

– Line 326: Is "navigational effectiveness" a standard term? If not, I wonder if a better term might be something like navigational cosine? The reason I ask is that if a larva has reached a desired temperature in a gradient then the "effective" thing to do might be to stop moving or move orthogonally to the gradient which would have a Navigational Effectiveness of 0.

We have rewritten effectiveness as “navigational efficiency,” which is a bit more accurate, describing how close to “straight line in the +x direction” navigation gets. Using <v_x>/<v> as the metric is consistent with the literature of larva crawling behavior. We like navigational cosine too (I describe it that way in talks sometimes), but will stay with the standard here if that is acceptable. It’s a good point about temperature gradients too, which is why in these experiments we always return animals to the center so they do not dwell at their desired temperature or edge of the arena – we mention this now in our justification for the pick-up/drop-off approach to extended experiments in the Introduction.

– Line 343: include details of statistical test.

Fixed, thank you.

Suggestions for improvement:– Include lines on the time series in Figure 4 to show the picking times.

Also added, thank you.

– In general, the data in Figure 4 is tantalizing, but because it's based on a single observation we can't rule out the increase in speed later in the recording isn't due to chance or some environmental change in the room. The most interesting thing would be to include some more data, but if that's not possible, including some more discussion of possible causes (including those that don't have anything to do with larval biology) in the text might help.

Additional experiments of this length are not doable at the moment, but there should be many more to come! We have added substantially to the Discussion section to talk about limitations of the experiments and changes in the surrounding laboratory environment. We don’t want to speculate too much with N=1 in this type of experiment.